# Diagnostic of Operation Conditions and Sensor Faults Using Machine Learning in Sucker-Rod Pumping Wells

**DOI:** 10.3390/s21134546

**Published:** 2021-07-02

**Authors:** João Nascimento, André Maitelli, Carla Maitelli, Anderson Cavalcanti

**Affiliations:** 1Federal Institute of Education, Science and Technology of Rio Grande do Norte (IFRN), Parnamirim 59143-455, Brazil; joao.nascimento@ifrn.edu.br; 2Department of Computer and Automation Engineering (DCA), Federal University of Rio Grande do Norte (UFRN), Natal 59078-970, Brazil; anderson@dca.ufrn.br; 3Department of Petroleum Engineering (DPET), Federal University of Rio Grande do Norte (UFRN), Natal 59078-970, Brazil; carlamaitelli@gmail.com

**Keywords:** sucker-rod pumping, machine learning algorithms, dynamometer card, petroleum industry

## Abstract

In sucker-rod pumping wells, due to the lack of an early diagnosis of operating condition or sensor faults, several problems can go unnoticed. These problems can increase downtime and production loss. In these wells, the diagnosis of operation conditions is carried out through downhole dynamometer cards, via pre-established patterns, with human visual effort in the operation centers. Starting with machine learning algorithms, several papers have been published on the subject, but it is still common to have doubts concerning the difficulty level of the dynamometer card classification task and best practices for solving the problem. In the search for answers to these questions, this work carried out sixty tests with more than 50,000 dynamometer cards from 38 wells in the Mossoró, RN, Brazil. In addition, it presented test results for three algorithms (decision tree, random forest and XGBoost), three descriptors (Fourier, wavelet and card load values), as well as pipelines provided by automated machine learning. Tests with and without the tuning of hypermeters, different levels of dataset balancing and various evaluation metrics were evaluated. The research shows that it is possible to detect sensor failures from dynamometer cards. Of the results that will be presented, 75% of the tests had an accuracy above 92% and the maximum accuracy was 99.84%.

## 1. Introduction

Wells produce oil through natural lift (flowing) when the reservoir has enough energy to lift fluids to the surface with commercially successful flow rates. However, when the reservoir pressure is not sufficient to overcome the sum of the pressure losses along the fluid path to the tanks, there is a need for the supplementation of energy to the reservoir through artificial lift methods. There are several methods of artificial lift and these are generally divided into two basic groups: pneumatic methods and pumping methods. Among the methods that use pumps, sucker-rod pumping is the oldest and the most used in the world. It is estimated that 90% of wells with artificial lift use sucker-rod pumping systems [1].

The main components of sucker-rod pumping wells are the prime mover, the pumping unit, the rod string and the downhole pump. The pumping unit transforms the rotary motion of the prime mover into the reciprocating motion necessary to operate the downhole pump [2]. The rod string connects the downhole pump with the pumping unit. The downhole pump works on the positive displacement principle and consists of a working barrel (cylinder) and plunger (piston). The plunger contains the discharge valve (traveling valve) and the working barrel contains the suction valve (standing valve). The two valves operate on the ball-and-seat principle and work like check valves [3]. The advantages of sucker-rod pumping systems are the simplicity of operation, low maintenance cost, durability of the equipment, operation range of flow and depth, good energy efficiency and the possibility of working with fluids of different compositions [4]. The basic components of a sucker-rod pumping well are shown in Figure 1.

In onshore oil fields, most wells produce oil by sucker-rod pumping systems and there are many wells for a few operators. From this perspective, the oil and gas industry has been investing in well automation to enable the monitoring of working conditions in a timely and accurate way. In general, the automation of wells operating by sucker-rod pumping systems consists of instrumentation, with a load cell and position sensors, a pump-off control device, a control panel and motor drive, and in some cases, a variable speed drive. Furthermore, a radio is required to receive and transmit information to/from SCADA (supervisory control and data acquisition) systems. Due to investments in digitizing field data, fiber optic networks are already found to be interconnecting wells and operating rooms.

Regarding the monitoring of sucker-rod pumping wells, the basic variables are: strokes per minute, daily pumping time, number of daily cycles, flow rate, well head or flow line pressure, operating status, operating mode, alerts, alarms and the surface dynamometer card. This card deserves a special mention. It is considered the main tool for analyzing the operation of wells with sucker-rod pumping systems [5]. It is a graphic that records the load and position values of the polished rod during a pumping cycle (basic period of the pump’s operation). The shape of the surface dynamometer card can reveal how the pumping system is operating [3]. However, as the sensors are on the surface, degenerative effects caused by the propagation of the load along the rod string change the shape of the card. This impairs the analysis on the surface, especially in deep wells [4]. For this reason and the difficulty of installing downhole sensors, Gibbs and Neely [5] developed the calculation of the dynamometer downhole card from the solution of the damped-wave equation.

Unlike the dynamometer surface card, the downhole card is better suited to reflect the pumping conditions, decreasing the chances of visual interpretation errors [6,7]. Regarding the shape of the cards, several factors influence the format: the operating conditions of the pump, the fluid capacity of the well, the type of fluid, the conditions of the valves, the existence of anchor in the column and gas interference [8]. In their book, Takacs [3] presents possible patterns for a dynamometer downhole card. Examples of patterns for operating conditions on the surface and downhole dynamometer cards are shown in Figure 2.

### 1.1. Sensor Faults in Sucker-Rod Pumping Wells

Sensors are essential components for automation systems with data acquisition and are widely used in various sectors [9] including in the oil industry. However, the sensors are also prone to faults due to their adverse operating environments. Thus, a quick diagnosis of the faults is important to prevent decreased production and increased downtime. When it comes to sensors in sucker-rod pumping systems, the load cell can be considered the most important sensor. It is constructed of stainless steel, it is hermetically selected (usually nitrogen gas filled) and installed in a polished rod. Its function is to measure the load values (fluid and rod string weight) during the pumping cycle.

There are different models of position sensors, however, due to the price and ease of installation, as it is common to use the position sensor switch, a mercury-wetted switch mounted on the base of the pumping unit and a magnet installed on the crank arm. Its operation produces a reliable dry contact closure once per pumping cycle. In this case, polished rod position is usually approximated by a simple sine function of pumping cycle time.

The union of data from the load cell and the position sensor allows the formation of the surface dynamometer card. In this case, it is possible to view some kinds of sensor faults on the dynamometer card itself. Examples of sensor faults in surface dynamometer cards: card with noise, line card, rotated card and card with inadequate load values. In addition, it is possible to mention the absence of cards caused by sensor faults.

The card with noise and the line card usually occur due to grounding fault or load cell cable connection fault, and the rotated card occurs due to an error in the position reference adjustment (the top of the stroke is generally used) and the card with inadequate load values occurs due to sensor error. Examples of sensor faults on the surface and downhole dynamometer cards are shown in Figure 3.

Despite the fact that most commercial controllers of sucker-rod pumping systems have alarms for sensor faults, some faults go unnoticed by the controllers, especially some types of line cards, noise cards, and mainly, rotated cards.

### 1.2. Diagnostic of Operation Conditions of Sucker-Rod Pumping Systems

Recognition and classification of patterns by visual similarity has become an important practice in the industry [10]. In the case of the oil and gas industry, which already has automated processes and a significant volume of data, the major challenge is to analyze a large amount of data. These data can be useful for insightful information and decision making. These enable better asset management [11]. In the case of the analysis of dynamometer cards of hundreds or thousands of wells in the same field, it is humanly unfeasible and ineffective to perform visual classification, in addition to requiring large experience. Therefore, it is essential to develop algorithms to identify events of interest, and consequently, improve asset management. In this perspective, machine learning (ML) guarantees the processing of a large volume of information in real time. It converts massive data into insights [11,12]. In essence, the application of ML facilitates the rapid identification of trends and patterns.

Recognition of dynamometer cards patterns is not a new procedure. Several studies have been written on the subject. In recent years, with the facilities inherent in ML algorithms and libraries, this number has been increasing. In 1990, Dickinson and Jennings [6] compared four pattern recognition methods. The methods were grid method, position-based Fourier descriptor, curvature-based Fourier descriptor and attributed-string-matching. In 1994, Nazi et al. [8] applied neural networks for downhole card diagnosis. After almost ten years, in 2003, Schnitman et al. [13] used relevant points from the cards and Euclidean distance to classify cards based on predefined patterns. In 2009, Souza et al. [14] used neural networks and card images to detect patterns. In 2011, Liu et al. [15] published results on the use of the AdaBNet and AdaDT algorithms for card classification.

In 2012, Lima et al. [4] evaluated the Fourier, centroid and K-curvature descriptors, via Euclidean distance and Pearson’s correlation. In 2013, Li et al. [16] used designated component analysis (DCA) and freeman chain, obtaining 95% accuracy. Again, Li et al. [17] achieved 98% accuracy, using four-point method, curve moments and support vector machine (SVM) with the error penalty adjusted via particle swarm optimization (PSO). Then, Wu et al. [18] used neural networks with back propagation (BP) from the card area, card perimeter, centroid and the area of the four corners of the card. Then, Yu et al. [19] compared the results for the following solutions: Fourier descriptors (FDs), geometric moment vector (GMV) and gray level matrix statistics (GLMX). Two years later, in 2015, Gal et al. [20] tested extreme learning machine (ELM) and feed forward and SVM neural networks; and Li et al. [21] proposed an unsupervised learning technique, fast black hole–spectral clustering (FBH–SC) for card classification.

In 2017, Zhao et al. [22] compared convolutional neural networks (CNN) from images and CNN from card data. He also compared the results with the random forest and K-nearest neighbors (KNN) algorithms. In 2018, Zheng and Gao [23] diagnosed downhole cards via decomposition and hidden Markov model; Zhang and Gao [1] used the fast discrete curvelet transform as dynamometer cards descriptors and sparse multi-graph regularized extreme learning machine (SMELM) as the algorithm; Zhou et al. [24] proposed a classification model based on Hessian-regularized weighted multi-view canonical correlation analysis and cosine nearest neighbor multi-classification for pattern detection; finally, Ren et al. [25] highlighted successful results when proposing root-mean-square error (RMSE) for card classification.

More recently, in 2019, Bangert and Sharaf [26] tested several ML descriptors and algorithms but highlighted that the best results came from the stochastic gradient-boosted decision tree, reaching more than 99% accuracy; Wang et al. [12] used deep learning (CNN) with 14 layers and 1.7 million neurons in cards treated as images of 200 by 100 pixels. It reached an accuracy above 90% in a real environment; Peng [27] generated and classified dynamometer cards from power curves (electric current) using CNN; Carpenter [28] brought together Siamese neural network (SNN), CNN, histogram of oriented gradients (HOG) and autoencoder neural network to generate an ensemble card classification model; and Sharaf et al. [29] used the card points, nine more characteristics from the controller and tested several models but cited that the best result was achieved with gradient boosting machines (GBM).

Abdalla et al. [30] used the elliptic Fourier and deep learning descriptors with some parameters optimized via genetic algorithms (GAs) and obtained an accuracy of 99.69%. Carpenter [31] tested eight ML models and presented their results from the highest accuracy to the lowest, as listed: gradient-boosted machines, extreme gradient-boosted trees (XGBoost), regularized logistic regression, random forest, light gradient-boosted trees, ExtraTrees, light gradient-boosted trees and TensorFlow deep-learning. In 2020, Cheng et al. [32] used transfer learning and SVM to automatically recognize working conditions in dynamometer cards.

The works cited tested several machine learning algorithms, including neural networks and deep learning, but did not reveal the difficulty level of the dynamometer card classification task. In reviewing the papers, it was observed that the most recent studies have an accuracy above 90%, regardless of the algorithm and techniques used. Therefore, it is necessary to know whether there is a standard in the works that allows to achieve these excellent results.

Unfortunately, these works did also not contribute to clarifying the differences with regard to the use of balanced and imbalanced datasets. A feature of the sucker-rod pumping system is that it has few faults, i.e., it is robust. This decreases the appearance of fault conditions, and consequently, the number of representative dynamometer cards. In addition, in oil fields with depleted reservoirs, the condition of the fluid pound is common. Under these conditions, the datasets recorded are very imbalanced and the largest number of dynamometer cards often occurs for the normal and fluid pound types. Likewise, several card-shaped features extractions or descriptors were also used in these works; however, the real impact of descriptors on the machine learning architecture for the diagnostic of operation conditions in sucker-rod pumping systems was not presented.

Therefore, despite the significant contributions of each work, doubts about the best algorithm, the best features extraction, imbalanced datasets, the amount of data necessary for efficient training of the models and the best tools for analyzing the results still exist [30]. In the search for answers to these questions, this paper presents results for several configurations, several ML algorithms, different balanced and imbalanced sets, the tuning of hyperparameters, and the use of automated machine learning (AutoML).

In this paper, real data from 38 sucker-rod pumping wells in the region of Mossoró, RN, Brazil, were used. More than 50,000 cards were classified by experts and distributed among eight modes of operation and two sensor faults common in this field. Sixty tests were carried out and divided into seven groups. The results of this research confirmed the feasibility of applying machine-learning to the diagnostic of operation conditions in sucker-rod pumping systems.

## 2. Materials and Methods

Due to social and economic changes, the oil and gas industry is facing unprecedented challenges. The excess of world supply, the development of electric cars, the pressures to improve their response to climate change and recently, the decrease in demand caused by the COVID-19 pandemic in 2020 and 2021, have required significant transformations [33]. The oil and gas industry needs to reduce costs and increase operational efficiency through the effective use of data, investing in technology to become more productive [34]. The use of machine learning can help in the improvement of technologies and processes, mainly where traditional engineering approaches were not efficient [35].

### 2.1. Machine Learning Essentials

Machine learning techniques extract knowledge from data to create models that can predict results from new inputs. The models are mathematical generalizations for probabilistic relationships between different variables [36]. Traditionally, the main applications treated by ML are regression and pattern recognition (classification). Regarding operation, regardless of the application, ML has a basic operation flow that is shown in Figure 4. The dataset can come from different sources, however, specialists must perform previous manipulations and they should use an algorithm to train the model that will later be used by applications to predict results. In the case of the low performance of the model, new training may occur, comprising new data and necessary adjustments.

For classification problems, whether binary or multi-class, the classifier’s performance is usually defined according to the confusion matrix, which is a tool capable of clearly presenting classifier errors. Its objective is to compare the set of predictions with real targets. The basic principle is to demonstrate the number of times instances of class A are classified as class B, and vice versa. For a binary classification problem, where a 2 × 2 matrix occurs, as shown in Figure 5, there are the following variables in the confusion matrix: TN (true negative), FP (false positive), FN (false negative) and TP (true positive).

In addition to the possibility of observing where the predictions went wrong in the confusion matrix, it is possible to obtain several metrics from the confusion matrix. Basic examples are: the precision case—it examines the proportion of positive predictions that are truly positive; the recall case—it measures the proportion of real positives correctly classified; and the accuracy case—it measures the proportion of real positives and negatives that are correctly classified. The metrics are shown in Equations (Equation 1)–(Equation 3):(1)$$precision=\frac{TP}{TP+FP}$$
(2)$$recall=\frac{TP}{TP+FN}$$
(3)$$accuracy=\frac{TP+TN}{TP+TN+FP+FN}$$

In ML projects applied in the industry, it is common to assume that positive classes are associated with faults and negative classes with normal operating conditions. Under these conditions, if the occurrence of false positives causes the improper halting of the process, resulting in losses for the industry, precision must be maximized. However, if the worst case is not to stop by a false negative, it is important to obtain a high recall [37].

In the context of this work, where dynamometer cards must be correctly classified, since solutions for faults can be delayed due to the cost of intervention, or due to the permanence of production viability, it is desired to have high precision and recall. In addition, the classification of normal cards is also important, as this condition may indicate the need for adjustments to increase production. Therefore, it is important to use metrics that combine precision and recall.

### 2.2. Imbalanced Datasets

Imbalanced datasets typically refer to a classification problem where the distribution of examples across the known classes is biased or skewed. Most machine learning algorithms assume that the number of objects in the classes is approximately similar. However, it is not possible to obtain balanced datasets under real operating conditions [37]. This represents a difficulty in ML algorithms, as they will be biased towards the majority group [38].

In the impossibility of collecting more data to balance the dataset, it is possible to change the dataset by sampling of instances. There are two basic ways of getting the same frequency for all classes: undersampling and oversampling [39]. With undersampling, instances of the majority class are excluded for classes where distribution is uniform. Oversampling adds copies of instances from the minor class for getting the same frequency for all classes. In this work, due to the size of the dataset and the difference in distribution between classes, the random oversampling technique was used, which randomly duplicates instances of minority classes to obtain a more balanced data distribution.

Another way to reduce the imbalance between classes is to generate synthetic samples. There are systematic algorithms to generate synthetic samples, one of them being the synthetic minority oversampling technique (SMOTE) [40]. SMOTE is an oversampling method that creates synthetic samples from the minor class instead of creating copies.

### 2.3. Metrics for Imbalanced Datasets

Many works have used the ratio between the number of correctly classified samples and the overall number of samples, i.e., accuracy Equation (Equation 3). However, in imbalanced datasets, the accuracy can provide an overoptimistic estimation of the classifier ability on the majority class [41,42]. In this work, $F_{1}score$ and Matthews correlation coefficient (MCC) calculated from the confusion matrices were the metrics adopted in the classification tasks.

The $F_{1}score$ corresponds to the harmonic average of precision and recall [43]. The harmonic mean gives greater weight to the lower values [44]. The $F_{1}score$ will only be high if the two measures, precision and recall, are high. An $F_{1}score$ metric is shown in Equation (Equation 4):(4)$$F_{1}score=\frac{2*precision\times recall}{precision+recall}$$

Unlike binary classification, multi-class classification generates an $F_{1}score$ for each class separately. However, if it is necessary to evaluate a single $F_{1}score$ for easier comparison, it is possible to use the $F_{1}macro$ [45]. The $F_{1}macro$ corresponds the averaging the $F_{1}score$ values for each class, according to Equation (Equation 5):(5)$$F_{1}macro=\frac{2}{N}\sum_{i=1}^{N}\frac{precision_{i}\times recall_{i}}{precision_{i}+recall_{i}}$$

The Matthews correlation coefficient, proposed by Brian Matthews in 1975 [46], is a metric based on statistical concepts for binary classification. MCC uses the idea of correlation commonly used in statistics to define the rate of a relationship between two variables [37]. Baldi et al. defines how to calculate the covariance values in terms of the values of the confusion matrix, according to Equation (Equation 6) [47]. The Matthews correlation coefficient is a discrete version of Pearson’s correlation coefficient and takes values in the interval [−1,1], where we have 1 if there is a complete correlation, 0 if there is no correlation and −1 if there is a negative correlation:(6)$$MCC=\frac{TP\times TN-FP\times FN}{\sqrt{\left(TP+FP)\times (TP+FN)\times (TN+FP)\times (TN+FN\right)}}$$

Similar to $F_{1}macro$, the multi-class problem can be evaluated with a single MCC metric [48]. The real data (X) and the prediction (Y) of the model are seen as statistical variables and *C* is the confusion matrix. In this case, MCC defines the rate of a relationship between variables, according to Equations (Equation 7)–(Equation 10) [49]:(7)$$MCC=\frac{cov(X,Y)}{\sqrt{cov(X,X)\times cov(Y,Y)}}$$
(8)$$cov\left(X,Y\right)=\sum_{i,j,k=1}^{\left|C\right|}\left(C_{ii}\times C_{kj}-C_{ji}\times C_{ik}\right)$$
(9)$$cov\left(X,X\right)=\sum_{i=1}^{\left|C\right|}\left[\left(\sum_{j=1}^{\left|C\right|}C_{ji}\right)\times \left(\sum_{k,l=1,k\neq 1}^{\left|C\right|}C_{lk}\right)\right]$$
(10)$$cov\left(Y,Y\right)=\sum_{i=1}^{\left|C\right|}\left[\left(\sum_{j=1}^{\left|C\right|}C_{ij}\right)\times \left(\sum_{k,l=1,k\neq 1}^{\left|C\right|}C_{kl}\right)\right]$$

### 2.4. Hyperparameter Tune

The machine learning algorithm’s performance is usually influenced by the choice of the values of its hyperparameters [50]. They are the variables that control model generalization and need a definition of values even before the training of the model is carried out. They represent characteristics used by algorithms when adjusting models, such as the maximum depth of a decision tree, the performance metric for regression algorithms and the number of neurons in a neural network [51]. Therefore, results from ML algorithms with tuned hyperparameters are often better than the standard configuration. On adjustment, the domain of a hyperparameter can be a real value, a binary or categorical value. For hyperparameters, the domains are mostly limited, with only a few exceptions [52]. In addition to the individual adjustment of hyperparameters, there is an adjustment by conditionality. That is, a hyperparameter can only be relevant if another hyperparameter assumes a certain value. For these cases, it is necessary to know the peculiarities of each algorithm.

As for tuning hyperparameters, there are several methods, but the most common are grid search, random search and Bayes search. Grid Search is the simplest of the methods. From a defined variable space for each hyperparameter, the algorithm tests all possible combinations [53]. In this case, it is possible to assume that this technique requires significant time, as the algorithm must test each possible combination to decide which one is best at the end of the process. The random search method, used in this work, performs a limited number of random combinations in the hyperparameter space to present the best performance in the outcome of the process [54]. Finally, the Bayes search method, unlike previous methods, which neglects preliminary promising results, builds a probabilistic model that replaces the optimization function of the problem [55]. This model selects promising hyperparameters. Then, they are tested in the actual objective function. With the test results, the model is updated. The procedures are repeated until the time limit or the maximum number of iterations is reached.

### 2.5. Machine Learning Algorithms

Regarding the algorithms used in this work, experiments were performed with decision tree [56], random forest [57] and XGBoost [58] algorithms. The choice of algorithms was motivated by the connection that exists between them that the three algorithms use a decision tree, evidently with significant changes in the structure. Furthermore, algorithms suggested by automated machine learning (AutoML) procedures were tested [52].

The decision tree algorithms build a decision model based on the values of the training instances. A decision tree consists of a root node, a number of non-terminal nodes of decision stages and a number of terminal nodes (final classifications) [59]. One of the advantages of the decision trees over other machine learning algorithms is how easy they make it to visualize data. In the case of the random forest and XGBoost algorithms, both are ensemble algorithms that use decision trees. They combine the decisions of various models to improve overall performance. This is important as the combination of several models can produce a more reliable forecast than a single model [60].

The random forest algorithm is an ensemble algorithm based on the application of bagging (bootstrap aggregating) on decision trees. The working principle is to randomly create decision trees. They are created only from some samples of the training data, but not from its totality [61]. Each tree will present its results. For classification problems, the most often presented results will be the chosen.

XGBoost (Extreme gradient boosting) is quite popular in ML challenges and is generally the first choice for most data science problems [62]. It belongs to a family of boosting algorithms and uses the gradient boosting (GBM) structure at its core. Boosting is a sequential and iterative technique. It adjusts the weight of an observation based on the last classification. If an observation was classified incorrectly, it tries to increase the weight of that observation and vice versa.

From raw data to the model ready for deployment, the concept of AutoML is linked to a set of libraries that provide a complete pipeline. Its applicability is important due to the existence of several algorithms and their various hyperparameters. This condition suggests an infinite number of possibilities, which certainly make it difficult to choose the algorithm and its hyperparameters [52]. Therefore, AutoML algorithms are those that indicate the best combination of algorithms and hyperparameters for a dataset in question. They aim at a better performance in prediction, and logically, ease of application. They can use Bayesian optimization, genetic algorithms and other optimization options.

As mentioned, AutoML can produce pipelines. In machine learning, pipelines are strategies for automating workflow. It is an automation sequence of tasks, including pre-processing, features extraction, model adjustment and validation stages. In this work, the TPOT (Tree-Based Pipeline Optimization Tool) library was used. TPOT is an open source genetic programming-based AutoML system that optimizes a series of feature pre-processors and machine learning models with the goal of maximizing classification accuracy on a supervised classification task [63].

### 2.6. Feature Extraction to Dynamometer Card

Machine learning involves the generation of models adjusted to a dataset to predict future observations or patterns. It is important to evaluate the relationship of the model with the features. Features are numerical representations derived from the available data and are linked to the model. Feature engineering is the process of formulating the most appropriate characteristics depending on the data, the model and the task [64]. The number of features is also important. If there are not enough descriptive features, the model is not able to serve the idealized purpose. However, if there are too many features, or irrelevant features, training the model is more expensive [64].

In image processing, when it comes to the description of shapes, case in point in this work, there are two options: the representation of the shape from external characteristics, i.e., its borders, or its internal characteristics. Considering that dynamometer cards are closed contours that can be treated as an image or sign, it is evident that the most appropriate form of representation is by contour [65].

In the description of shapes based on the contour, in a sequence of ordered pairs $\left(x_{0},y_{0}\right),\left(x_{1},y_{1}\right),\left(x_{2},y_{2}\right)$$,\ldots ,(x_{N-1},y_{N-1})$, each ordered pair can be treated as a complex number Equation (Equation 11):(11)s(k)=x(k)+jy(k)
where k=0,1,2,…,N−1 and *N* is the number of points in the contour. In this case, the *x* axis is treated as the real axis and the *y* axis is an imaginary axis of a sequence of complex numbers. This kind of representation has an advantage because it reduces a 2-D problem to a 1-D problem. Another common variation for the representation of shapes arises from the calculation of the centroid Equations (Equation 12) and (Equation 13):(12)$$x_{c}=\frac{1}{N}\sum_{i=0}^{N-1}x_{i}$$
(13)$$y_{c}=\frac{1}{N}\sum_{i=0}^{N-1}y_{i}$$

After calculating the centroid, it is possible to use $x_{c}$ and $y_{c}$ to create a new contour representation, according to Equation (Equation 14):(14)$$z\left(k\right)=\left(x_{k}-x_{c}\right)+j\left(y_{k}-y_{c}\right)$$

A common boundary-based shape descriptor is the Fourier descriptor [10]. With this representation Equation (Equation 14), from the discrete Fourier transform (DFT), it is possible to calculate the contour Fourier descriptors with Equation (Equation 15):(15)$$a\left(k\right)=\frac{1}{N}\sum_{k=0}^{N-1}z\left(k\right)e^{-\frac{j2\pi nk}{N}}$$
where n=0,1,2,⋯,N−1 and a(k) are the Fourier coefficients of the contour. Regarding the rotation invariance, the descriptors are invariant if the magnitude of the transformation is used |a(k)| [10]. However, in the case of dynamometer cards, it is not important to consider the rotation invariance since some types of cards can be modified after rotation. It is the case of the standing valve that there are leakage cards in the traveling valve. Once in the frequency domain, peaks will indicate the frequencies that most occur in the signal. In this case, the larger and clearer a peak is, the more prevalent the frequency in the signal will be. For the use of the Fourier descriptors in ML, the full use of the descriptors or just the location and height of the peaks in the frequency spectrum for training classifiers can be considered. In this work, the magnitude vector p(k) resulting from the Fourier transform calculation was used Equation (Equation 15). The result is seen in Figure 6:(16)p(k)=|a(k)|

The wavelet transform became an active area of research for multi-resolution signal and images analysis [66]. Thus, an interesting alternative for analyzing dynamic signals or images is the wavelet transform instead of the Fourier transform. Unlike the Fourier transform, whose base functions are sinusoidal, wavelet transform is based on small waves called wavelets. They have varying frequency and limited duration [65]. The basic objective of wavelet functions is the hierarchical decomposition of signals. It is the representation of the signal or image at different levels of resolution or scales [67]. There are several wavelet functions that differ in shape, smoothness and compression. A wavelet function must be selected based on the best adaptability to the feature that one wishes to observe in the signal or image. As for the transformations, there is the continuous wavelet transform (CWT) and the discrete wavelet transform (DWT). In the context of this work, the continuous wavelet transform is described as Equation (Equation 17):(17)$$C_{w}\left(s,b\right)=\frac{1}{\sqrt{\left|s\right|}}\int z\left(k\right)\psi \left(\frac{k-b}{s}\right)dk$$
where ψ is the wavelet function, *s* is the scale factor and *b* is the translation factor. The wavelet coefficients $C_{w}$ are defined for b=0,1,2,…,N−1. Basically, what differentiates the continuous from the discrete wavelet transform is the use of continuous values for the scale and translation factors, in the case of CWT, and discrete values for DWT. Particularly, for the DWT case, the scale factor increases by powers of two (s=1,2,4,…), whereas the translation factor must increase sequentially (b=0,1,2,3,…).

The use of wavelet descriptors in ML involves the choice of the transformation mode (CWT or DWT), the type of wavelet function and the choice of the level of decomposition (scale factor). This variety of choices can be considered as another advantage of transformed wavelets. However, even with some recommendations for using the wavelet functions in the literature [68], testing is necessary to achieve the best choice.

In this work, tests were carried out with continuous and discrete transforms for different Wavelet Functions. Initially, the use of CWT coefficients for a single scale was tested. An example of CWT application with only a scale on a normalized downhole dynamometer card is shown in Figure 7.

Regarding DWT, these are often implemented as filter banks. It is like a cascade of low-pass and high-pass filters allowing the division of signals into several frequency sub-bands. After applying the transform and having the approximation and detail coefficients, it is possible to choose how these coefficients will be used as inputs by ML algorithms. The options can vary, and they depend on the signal evaluated. The choice can either privilege the results of the low-pass filter, the high-pass filter or the combination of the two. It is also important to pay attention to the level of decomposition of the filter. An example of applying DWT to a normalized downhole dynamometer card is shown in Figure 8.

### 2.7. Methodology

In general, the application of ML for solving classification problems is divided into two phases: experimentation and operationalization [69]. ML experimentation refers to efforts focused on observing the problem; obtaining, exploring and preparing the data; selecting the algorithm; and model validation [69,70]. Operationalization of ML refers to the process of implementing models, consuming, and monitoring results, in an efficient and measurable way [69]. Figure 9 shows the ML experimentation steps used in the development of this work.

Regarding the stages of the experimentation phase, the observation of the problem is extremely important for the success of the project. At this stage, one must look at how the problem is being treated today, what operations conditions and sensor faults occur most frequently, what treatment is given after the detection of the operation conditions, how the problem is expected to be solved and mainly, how the results will be applied to generate value for the business [70].

In the stage of obtaining the data, it is necessary to know the source of the data, the interface for accessing the data, observe sample rates, convert the data to a better handling format and start the manual classification process with the help of specialists [70]. This work developed a tool that accesses a PostgreSQL database and facilitates the manual classification of the cards for it allows for easy and quick navigation between wells and their cards. In the application, the selected surface cards are presented with their respective downhole cards and guidelines. They indicate the quality of the data coming from the load sensors in relation to the well design. Some cards can be ignored if the card position in the graph is not well aligned with the design guidelines. Because the dynamics are slow and the average rate of card acquisition per well is 10 min, the cards are sequentially very similar. In this case, the change in operating conditions does not occur suddenly. This characteristic of the wells facilitates the manual classification process. The classification and selection scheme of background cards used in this work are shown in Figure 10.

After classification, the application saves the already normalized downhole cards, the data from the well and information from the card itself. The normalization procedure allows downhole cards from different wells to be treated with the same scale [30]. This is essential for the proper functioning of ML algorithms. For normalization, in addition to the standardization of 100 points per downhole card, the following Equations (Equation 18) and (Equation 19) were used:(18)$$x_{i_{normalized}}=\frac{x_{i}-x_{min}}{x_{max}-x_{min}}$$
(19)$$y_{i_{normalized}}=\frac{y_{i}-y_{min}}{y_{max}-y_{min}}$$
where $x_{i}$ is the plunger displacement that will be normalized, $y_{i}$ the plunger load value that will be normalized, $x_{min}$ and $x_{max}$ are, respectively, the minimum and maximum plunger displacement and $y_{min}$ and $y_{max}$ are, respectively, the minimum and maximum plunger loads.

Regarding the data exploration stage, it is necessary to study the data that make up the dataset, check the possibilities of correlation between the data and visualize the dataset. In this work, after the manual classification of the cards, the following number of cards was obtained by operation conditions or sensor faults. Table 1 shows the distribution of cards by type used in this work.

It is observed that the number of cards by type is poorly distributed, making the dataset imbalanced. This problem, if not observed, can affect the classification results. In this case, there are undersampling and oversampling techniques to minimize the problem. The ideal case is that many samples for each type and a balanced dataset regarding the distribution by types. A solution adopted by some papers [13,14,22] is the use of cards designed to compose a balanced dataset. However, the random oversampling technique was used in this work.

Now, it is necessary to prepare the dataset to run the ML algorithms. In this stage, tasks such as the exclusion of spurious data, separation of training and test sets, data standardization, and mainly, the feature engineering process should be performed. In the context of this work, where dynamometer cards with hundreds of points are treated, dimensionality reduction procedures can also be performed, using, for example, principal component analysis (PCA) [71].

In ML projects, before selecting the algorithm and models to train, good practice recommends dividing the dataset into two sets: the training set and the test set. It is common to use 80% of the data for training and keep 20% for testing [70]. However, in classification problems where the dataset is imbalanced, it is important to consider the stratified separation. It is necessary to keep the proportions chosen and representative for each class.

In the next step—selecting the algorithm—a good procedure is to start testing on simpler algorithms, then move on to more complex algorithms. In addition, it may be interesting to perform the first tests with a smaller dataset, favoring the possibility of the quick exchange of algorithms. It is also recommended to test the algorithms without previous configurations, and later with hyperparameter tuning. Thus, it is possible to evaluate the effect of hyperparameters and whether the adjustments thus favored greater accuracy. After the tests, some algorithms with better performance will be chosen. It is important that these algorithms are tested in larger datasets to prove efficiency, and if necessary, to implement new adjustments. Finally, the use of AutoML should be considered due to its ability to identify the pipelines, algorithms and hyperparameters that are most suitable for the problem in question.

Along with the choice of algorithms, there is the performance evaluation of the classifiers. In this case, it is a good starting point to compare accuracy with null accuracy. The null accuracy corresponds to the percentage of the class with the most samples in the dataset. In the case of this work, the class of fluid pound cards has a higher number of instances, corresponding to 76%. If any algorithm classifies all cards as fluid pound, it might still be considered a good performance. This indicates that the starting point for evaluating the metrics is 76% accuracy. In addition to accuracy, it is recommended to examine the confusion matrices to see where the classifier is going wrong. In the problem in question, however, it is very common to find classification errors involving cards such as the fluid pound and gas interference. Sometimes, there are errors with normal cards—that almost fill up the barrel with fluid pound. In the latter case, at the time of classification via specialist, it is interesting to establish the maximum level, in terms of effective stroke at the bottom of the well, which will be considered as fluid pound. In addition, after implementation, it is suggested that the classifier does not provide exact answers for users of the system, but rather, the probability of correspondence for the types evaluated. This helps the user to take more assertive measures.

Finally, it is necessary to examine the metrics that are extracted from the confusion matrix, and more importantly, evaluate the selected models with cards from other wells that are not in the dataset. This allows cards from different wells and with different dynamics to be classified and submitted for evaluation. This procedure can guarantee that the selected model is performing well or if it still needs adjustments.

### 2.8. Implementation of Experiments

To implement the experiments used in this research, the basic structure of steps and procedures presented in Figure 11 was followed. This figure demonstrates that, at the beginning of the experiment, the PostgreSQL database is consulted, where the previously classified background cards are stored. The result of the query is transformed into a structure of the Pandas package, called dataframe. Subsequently, the data are converted into a more suitable form. At this point, some analyses can be performed, such as checking the correlation between variables. With the vectors that represent the dynamometer cards, the descriptor calculation procedures are performed, including the calculation of centroids as well as the Fourier and wavelets descriptors. After the previous step, which lasts a few minutes, the cards are mixed in a stratified way to generate the training and test datasets. Depending on the test to be performed, balancing procedures of the training dataset are applied. Then, the ML algorithms are applied to generate models, and finally, reports are generated containing the metrics that must be analyzed. These tests were performed using scikit-learn [72], PyCM [73], imbalanced-learn [74], TPOT [63], PyWavelets [75] and Pandas.

## 3. Results

This section was split into four subsections for the sake of clarity. First, the experimentation procedures are presented; in the second section, the results obtained from models trained using small datasets (30, 90 and 180 instances) are presented; in the third, the results of models trained from a large dataset (40,078 instances) are outlined; and in the last section, the model with the best result and general conclusions are presented.

### 3.1. Experimentation Procedures

Regarding the execution of the experiments, the descriptors used were: Fourier descriptors (with DFT) and wavelet descriptors (with CWT and mother wavelet of the Mexican hat type). In addition, the normalized load values from the downhole card were also used. The algorithms used decision tree, random forest and XGBoost, as well as pipelines suggested by AutoML procedures. Regarding the metrics, accuracy, precision, recall, $F_{1}macro$ and MCC were evaluated. In order to standardize the name of the resources used in the tests, Table 2 summarizes the names and tags used for each experiment.

### 3.2. Results from Models Trained by Small Datasets

Experiments were carried out with 24 models trained from datasets with 30, 90 and 180 instances equally split between classes. The tests, on the other hand, were carried out on a dataset with 50,098 instances. The purpose of the tests was to assess the minimum number of instances to achieve good accuracy and to determine the impact of changing algorithms and descriptors on the classification process.

The tests were organized into four groups (A, B, C and D). Each group has characteristics that vary according to the techniques employed and the number of instances per class. Table 3 shows the organization of the tests into groups.

In group A, all classes in the training set had 30 instances. The tests were performed with all descriptors (Fourier, wavelet and loads) and the three algorithms (decision tree, random forest and XgBoost), without the tuning of hyperparameters. The metric values (accuracy, $F_{1}macro$ and MCC) of the tests performed in group A are shown in Figure 12. All graphs listed in the subsequent sections are normalized.

In group B, all classes in the training set had 30 instances. The tests were performed with all descriptors (Fourier, wavelet and loads) and pipelines suggested by AutoML procedures by TPOT. The metric values (accuracy, $F_{1}macro$ and MCC) of the tests performed in group B are shown in Figure 13. About the generated pipelines, they will be presented in Table 4. In this case, the use of AutoML did not help to improve the results, showing that the number of training instances is still low. Regarding the genetic algorithm used by TPOT, 15 generations and a population of size 10 were used.

In group C, all classes in the training set had 90 instances. The tests were performed with all descriptors (Fourier, wavelet and loads) and the three algorithms (decision tree, random forest and XgBoost). The metric values (accuracy, $F_{1}macro$ and MCC) of the tests performed in group C are shown in Figure 14. In this case, a maximum accuracy of 97.90% was observed. This was already considered an excellent result when compared to other works.

In group D, all classes in the training set had 180 instances. The tests were performed with all descriptors (Fourier, wavelet and loads) and just an algorithm (decision tree). The metric values (accuracy, $F_{1}macro$ and MCC) of the tests performed in group D are shown in Figure 15. The objective of the tests in group D was to evaluate the simplest algorithm used in this work (decision tree) with a slightly higher number of instances.

Multi-class classification problems are usually divided into a set of binary problems [82]. There are several ways to carry out the binary decomposition: one-versus-all (OvA), one-versus-one (OvO) and ECOC (error correcting output codes) [83]. In this work, the metrics were calculated using OvA. The confusion matrix of group A that presented the best accuracy (86.00%) is shown in Figure 16.

The confusion matrix of group D that presented the best accuracy (98.17%) is shown in Figure 17.

In the confusion matrix in Figure 16, the zero-oneloss (number of misclassified instances) [84] was 7011. Already in the confusion matrix in Figure 17, the zero-oneloss is 915. This represents a significant reduction in misclassified instances, even with a small number of training instances per class. As observed in the confusion matrices and graphs presented in this section, the results showed excellent accuracy. Errors occurred more frequently (FN and FP) for the normal and fluid pound classes.

### 3.3. Results from Models Trained by Large Datasets

Despite obtaining results with acceptable accuracy in the previous tests, having a larger training set makes it possible to improve the performance of the models. After all, the greater the number of instances, the easier it is for the model to correctly generalize. This dataset has 40,078 instances distributed according to Table 5.

Experiments were carried out with 36 models trained from the dataset with 40,078 instances and organized into three groups (E, F, and G). In addition to the results of models trained with more instances, the tests sought to present the effects of balancing and tuning hyperparameters. The balancing technique used was random oversampling. Table 6 shows the organization of the tests in groups.

In group E, the tests were performed with all descriptors (Fourier, wavelet and loads), the three algorithms (decision tree, random forest and XgBoost) and balanced/imbalanced datasets. Hyperparameter tune was not used. The best result, in terms of accuracy, was 99.81% for the test with loads with XgBoost and balanced dataset. The metric values (accuracy, $F_{1}macro$ and MCC) of the tests performed in group E are shown in Figure 18.

In group F, the objective was to observe the effects of hyperparameter tuning on the results. The tests were performed with all descriptors (Fourier, wavelet and loads), the three algorithms (decision tree, random forest and XgBoost) and balanced/imbalanced datasets. The best result, in terms of accuracy, was 99.82% for the test with loads with XgBoost and balanced dataset and hyperparameter tune. The metric values (accuracy, $F_{1}macro$ and MCC) of the tests performed in group F are shown in Figure 19.

The tests in group G were performed with all descriptors (Fourier, wavelet and loads), balanced/imbalanced datasets and pipelines suggested by AutoML procedures by TPOT. The metric values (accuracy, $F_{1}macro$ and MCC) of the tests performed in group G are shown in Figure 20.

About the generated pipelines, they will be presented in Table 7. Regarding the genetic algorithm used by TPOT, 15 generations and a population of size 5 were used.

Figure 21 summarizes the metrics for groups E, F and G. In this case, it can be seen that any of the results are acceptable.

Figure 22 shows a histogram of the accuracy of groups E, F and G. This shows that, regardless of the algorithm, the descriptors and balanced state of the dataset, the result is considered excellent, considering other works.

### 3.4. The Best Model for Diagnostic of Operation Conditions and Sensor Fault

The best result occurred with the model that used a pipeline proposed by the AutoML procedure. The model was trained with the following pipeline: MultinomialNB, PCA and KNeighborsClassifier. The accuracy value was 99.84% and the value of zero-oneloss was 16. The model’s confusion matrix is shown in Figure 23.

A histogram with the accuracy values for all tests is shown in Figure 24. It shows that, regardless of the algorithm, the descriptors and the state of balance, the models generalize well and the results are acceptable.

As shown in Figure 25 and Figure 26, $F_{1}macro$ is the most suitable metric to evaluate the classification performance of downhole cards, since the MCC did not show great variations.

## 4. Discussion

Regarding the bad distribution of cards in the dataset, the high number of fluid pound cards is common in mature fields and in many cases, it is considered a normal condition due to depleted reservoirs.

With the help of specialists, it is possible to identify that some modes of operation require greater attention, either due to similarity with other modes, or the need to evaluate more resources. In such cases, ensuring the classification result will only be possible after investigative procedures. For example, the shape of the rod parted can easily be confused with the top of the fluid level conditions. Another example is the possibility of similarity between the operation fluid pound and gas interference. In both cases, the barrel is partially filled with fluid (water and oil). However, in the case of gas interference, due to the lack of an efficient separator, the barrel is filled with gas, generating pumping problems. The gas is highly compressible, and it makes it difficult for the pump valves to function. This problem has been reported by other works [4,5,6,7,8,9,10,11,12,13,14,15,16,17,18,19,20,21,22].

The results obtained by the load values may assume that for the downhole card classification problem, machine learning does not require descriptors. The load values of the cards are already an excellent descriptor as long as these are normalized.

As mentioned before, it has been a long time since the oil industry invested in the automation of wells. Some automation equipment in the field is considered old. However, the advances in new controllers or instruments still do not justify the replacement, except for communication systems. They tend to be replaced by optical fiber systems. Many of the problems also occur due to maintenance failures. Therefore, the problem observation phase is important for detecting these failures.

When implementing the solution, it is recommended that online solutions are chosen where the system user can collaborate with the adjustment of the model indicating classification errors. However, for proper functioning, it is important to define a pipeline that assesses whether the new model adjustment has improved performance or not. If not, the pipeline would return to its previous condition.

This work did not directly compare results with other works, as it understands that the comparison is unfair since the datasets are different.

## 5. Conclusions

This work presented results for the application of machine learning in the detection of operating conditions and sensor faults in sucker-rod pumping systems. To classify the cards, three algorithms were tested: decision tree, random forest and XGBosst. In addition, procedures for tuning hyperparameters and pipelines generated by automated machine learning (AutoML) were tested. The results demonstrated that from a training dataset with many instances, good ML algorithms can produce high accuracy rates in the classification process. However, even with few training instances, but with good algorithms, the results are satisfactory.

This study also tested different descriptors (features), but specifically Fourier descriptors with centroid and wavelets descriptors with centroid. However, tests without descriptors showed results very similar to the other descriptors. This reinforces that good machine learning algorithms are sufficient to achieve good results.

Regarding the metrics, the results showed that $F_{1}macro$ is more suitable for performance evaluation of the models, since the MCC did not show sensitivity to the errors presented in the confusion matrix.

Concerning the state of balance of the training sets, no major differences were observed in the results, even though the average accuracy values of the balanced tests (99.74%) of group E have been better than the average of the imbalanced tests of the same group (99.68%). Even with the good quality of the results, the analysis of techniques of the synthetic generation of instances to balance the datasets can still be interesting, as this would be a case of testing SMOTE or something similar.

AutoML proved to be very effective for solving the card classification problem. The generated pipelines incorporated characteristic normalization procedures, including PCA. The most suggested algorithms were k-nearest neighbors and the extra-trees classifier. The great advantage observed is in the fact that the pipeline is complete (preparation + standardization + algorithm), saving a lot of project time.

For further research, it is recommended to use the downhole card classification feature associated with production loss assessments and decision support systems. Currently, this is necessary because automated wells offer hundreds or thousands of cards per day. Therefore, except for standards that indicate serious equipment failures, the other standards allow the well to continue producing. However, it is necessary to identify losses of production to make quick and effective decisions.

## Figures and Tables

**Figure 1 sensors-21-04546-f001:**
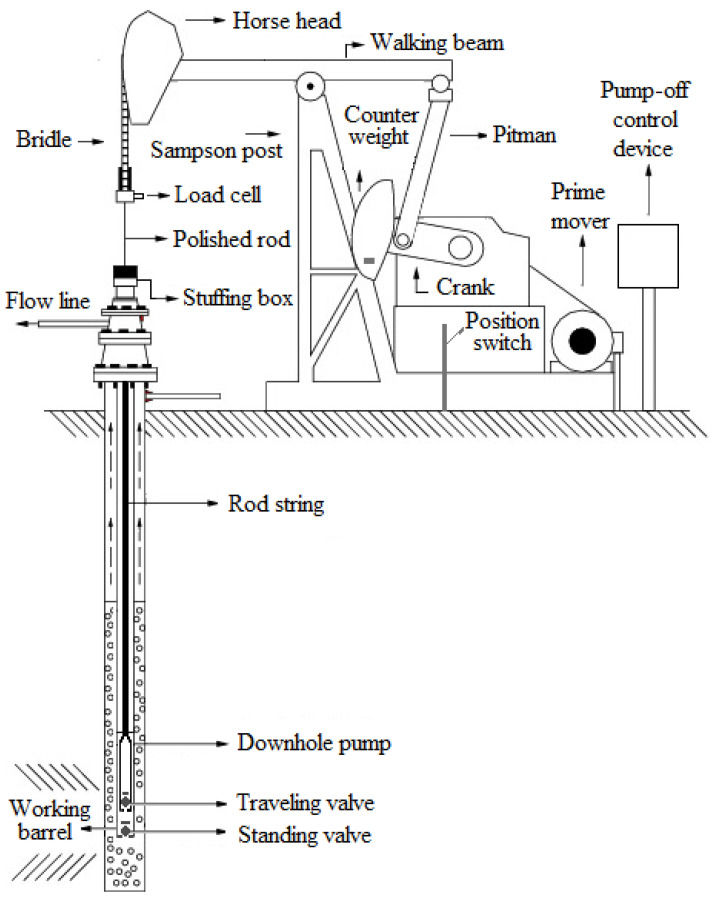
Basic components of a sucker-rod pumping well.

**Figure 2 sensors-21-04546-f002:**
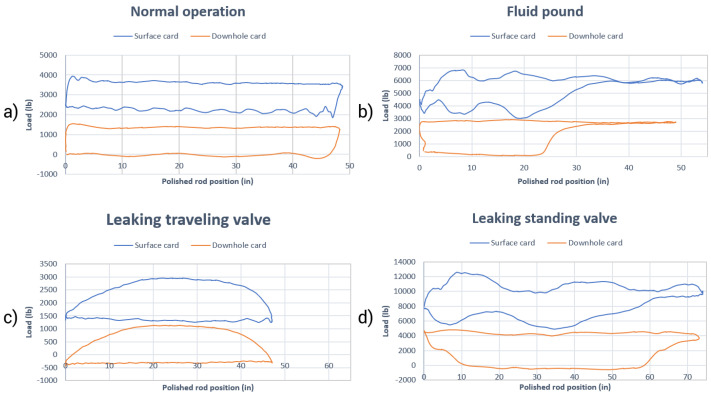
Examples of patterns for operating conditions on the surface and downhole dynamometer cards: (**a**) normal operation; (**b**) fluid pound; (**c**) leaking traveling valve; and (**d**) leaking standing valve.

**Figure 3 sensors-21-04546-f003:**
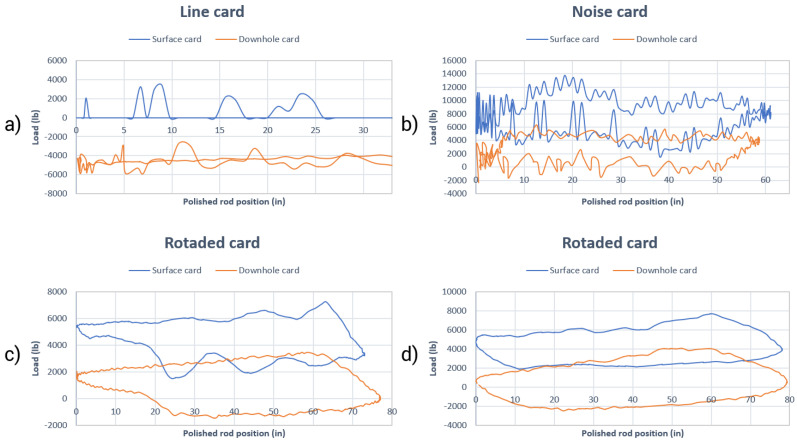
Examples of sensor faults in surface and downhole dynamometer cards; (**a**) line card; (**b**) noise card; (**c**,**d**) rotated card.

**Figure 4 sensors-21-04546-f004:**
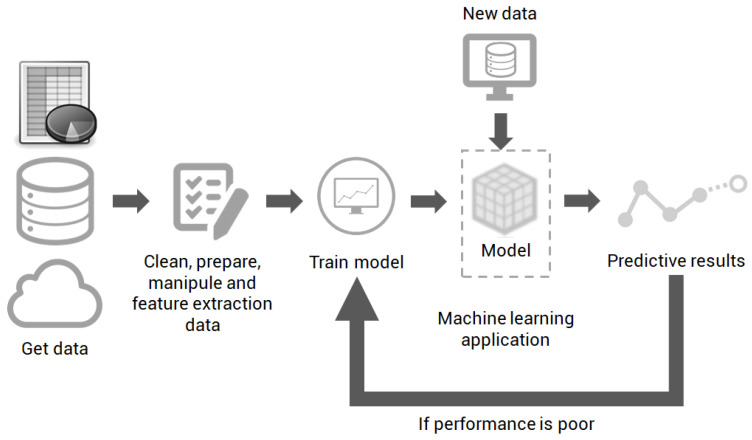
Basic flow of operation of a project with machine learning.

**Figure 5 sensors-21-04546-f005:**
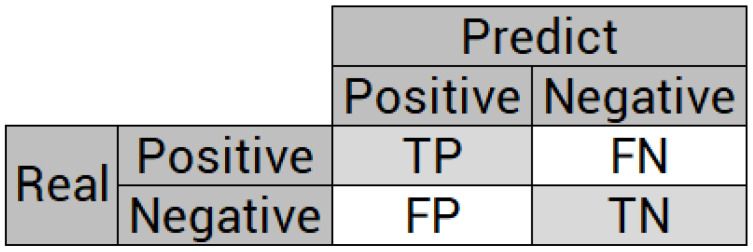
Confusion matrix for binary classification problems.

**Figure 6 sensors-21-04546-f006:**
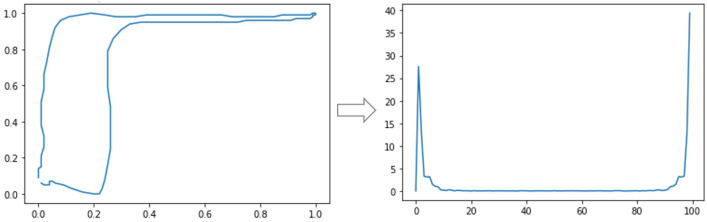
Application of the Fourier transform in a normalized downhole dynamometer card using Equations (Equation 14)–(Equation 16).

**Figure 7 sensors-21-04546-f007:**
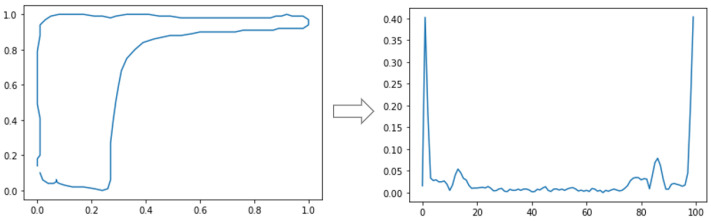
Application of the CWT, with only one scale, in a normalized downhole dynamometer card. Equations (Equation 14) and (Equation 17) and the magnitude of the wavelet coefficients were used.

**Figure 8 sensors-21-04546-f008:**
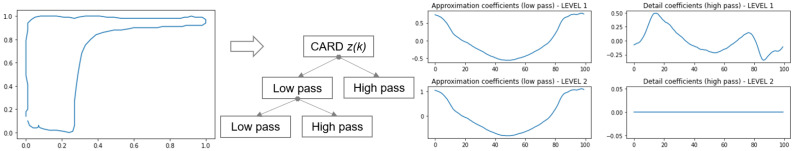
Application of DWT on a normalized downhole dynamometer card. Equations (Equation 14) and (Equation 17) and discrete values for *s* and *b* were used.

**Figure 9 sensors-21-04546-f009:**
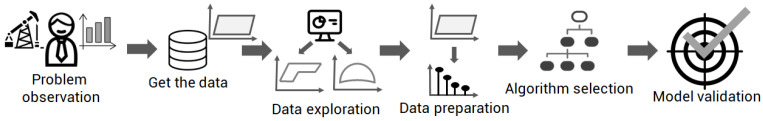
Common steps for experimenting with a machine learning project. These steps were used in the development of this work.

**Figure 10 sensors-21-04546-f010:**
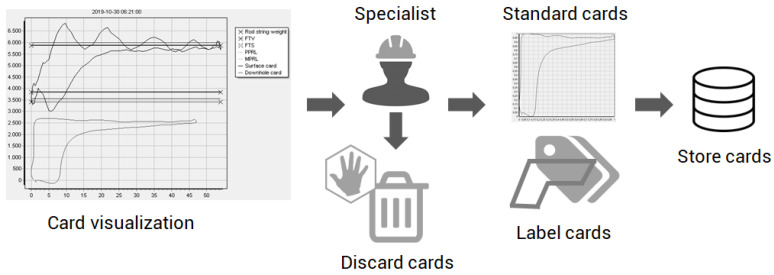
Scheme of classification and manual selection of downhole cards used in this work.

**Figure 11 sensors-21-04546-f011:**
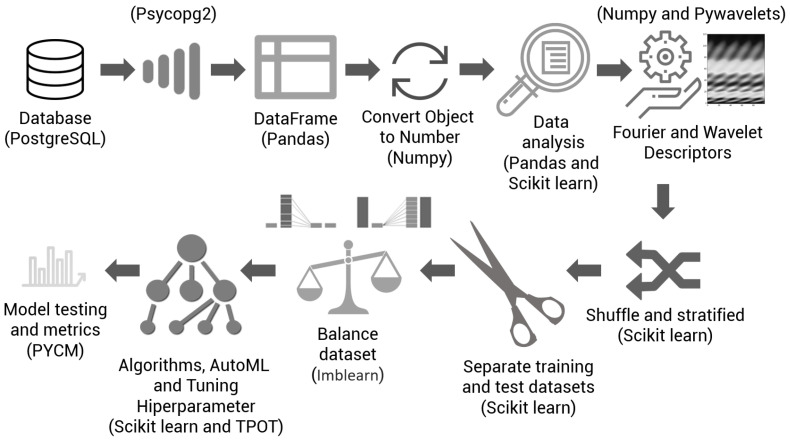
Implementation of experiments.

**Figure 12 sensors-21-04546-f012:**
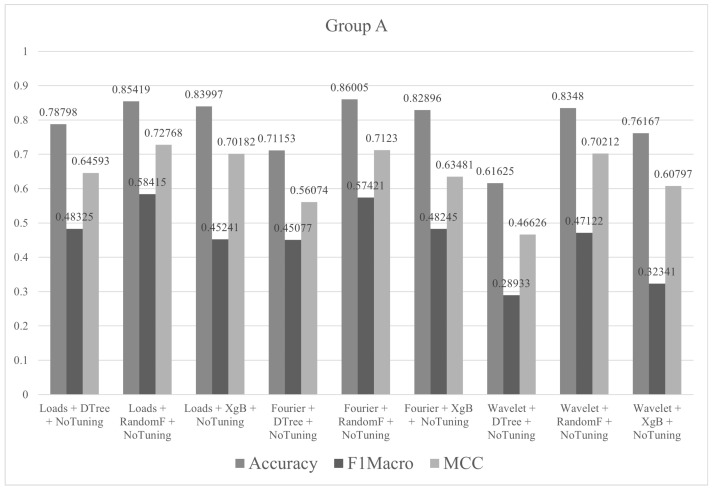
Metrics of group A.

**Figure 13 sensors-21-04546-f013:**
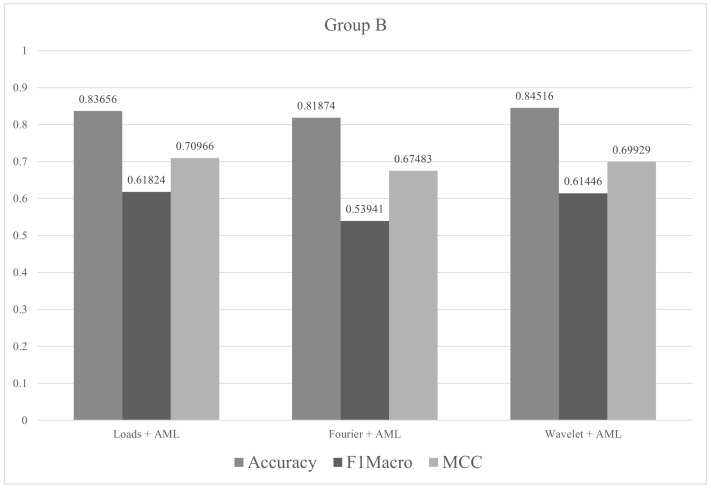
Metrics of group B.

**Figure 14 sensors-21-04546-f014:**
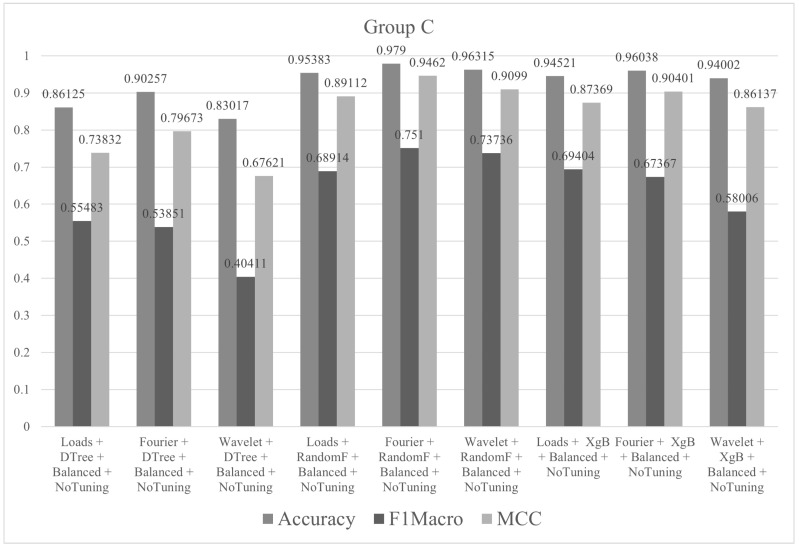
Metrics of group C.

**Figure 15 sensors-21-04546-f015:**
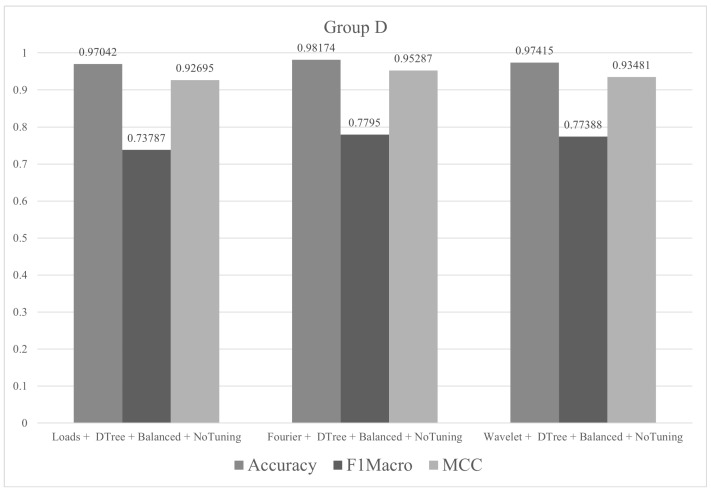
Metrics of group D.

**Figure 16 sensors-21-04546-f016:**
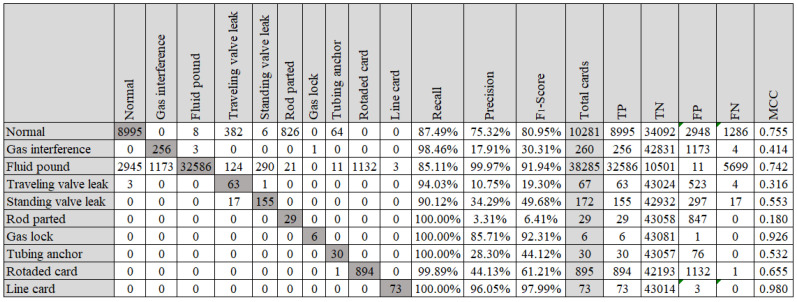
Confusion matrix for training dataset with 30 instances per class (Fourier descriptor, random forest and no hyperparameter tuning).

**Figure 17 sensors-21-04546-f017:**
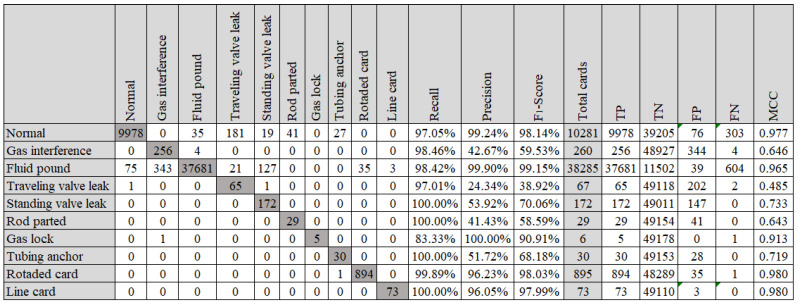
Confusion matrix to training dataset with 180 instances per class (Fourier descriptor, decision tree and no hyperparameter tuning).

**Figure 18 sensors-21-04546-f018:**
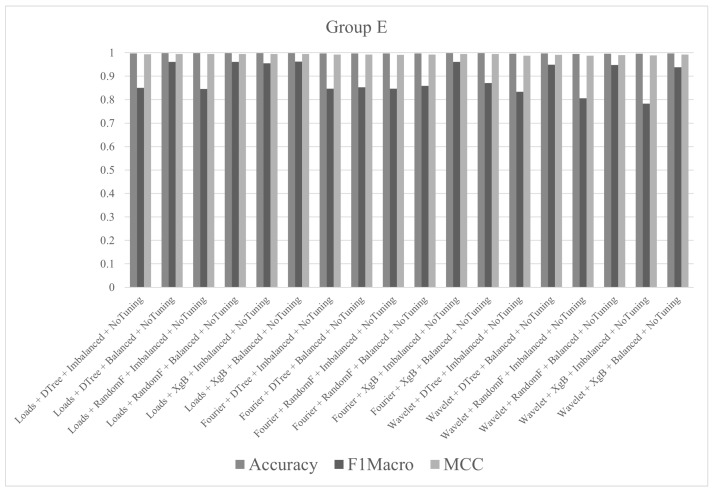
Metrics of group E.

**Figure 19 sensors-21-04546-f019:**
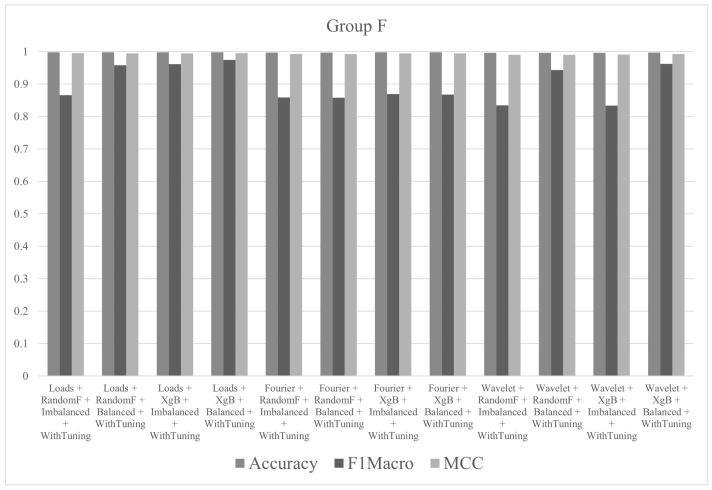
Metrics of group F.

**Figure 20 sensors-21-04546-f020:**
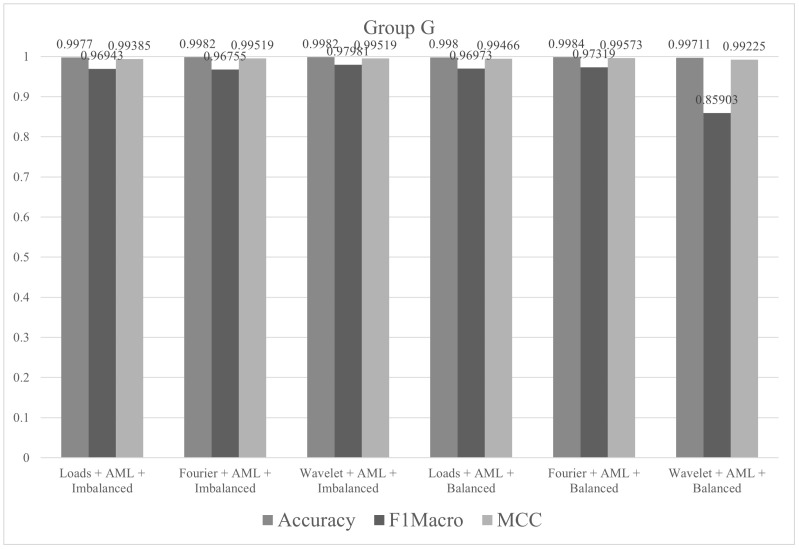
Metrics of group G.

**Figure 21 sensors-21-04546-f021:**
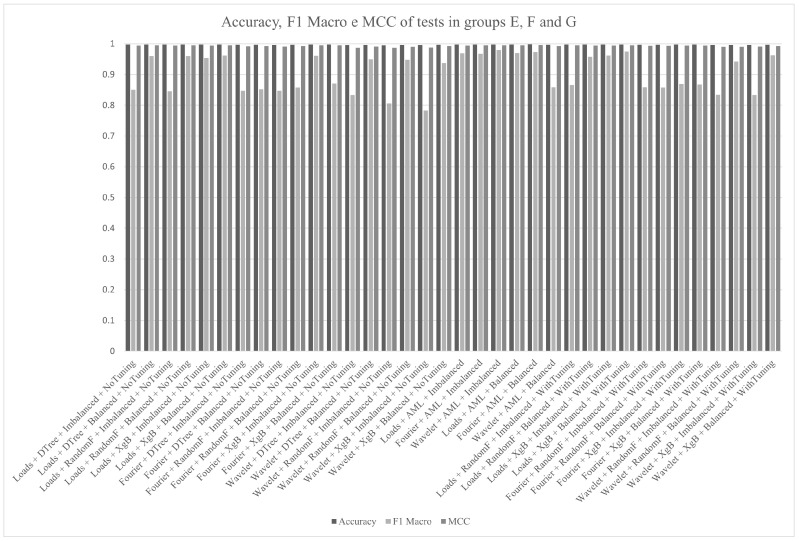
Accuracy, $F_{1}macro$ and MCC of the groups E, F and G.

**Figure 22 sensors-21-04546-f022:**
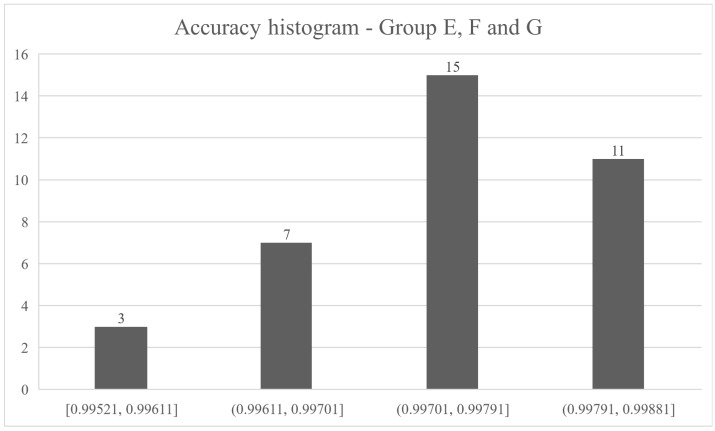
Accuracy histogram of the groups E, F and G.

**Figure 23 sensors-21-04546-f023:**
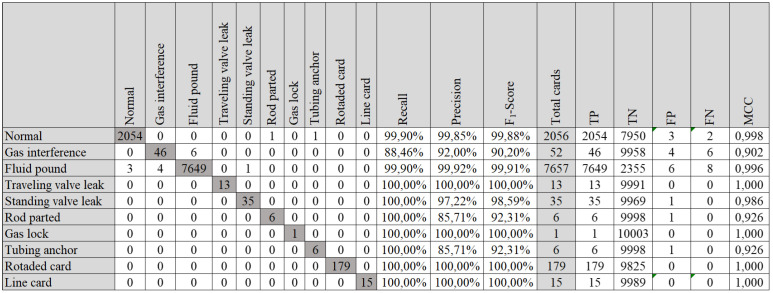
Best confusion matrix of this work (Fourier descriptor, balanced and AML)—(MultinomialNB, PCA and KNeighborsClassifier)—group G.

**Figure 24 sensors-21-04546-f024:**
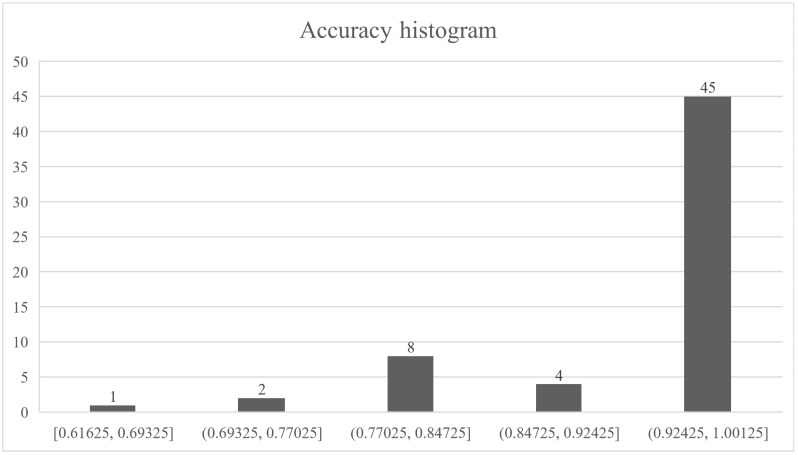
Accuracy histogram.

**Figure 25 sensors-21-04546-f025:**
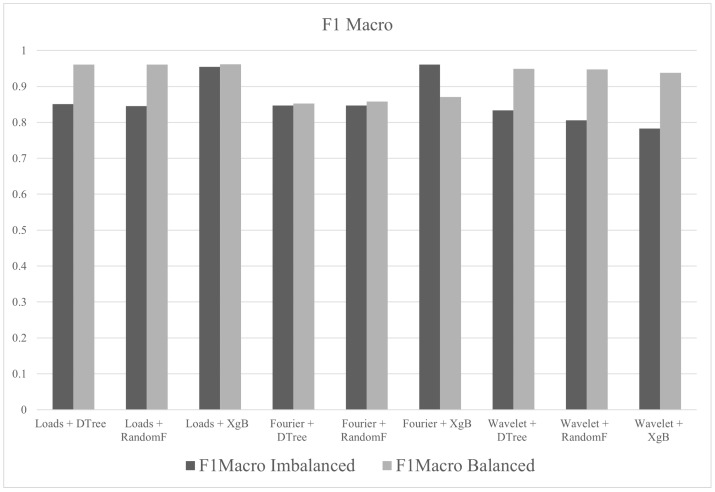
$F_{1}macro$ for balanced and unbalanced datasets.

**Figure 26 sensors-21-04546-f026:**
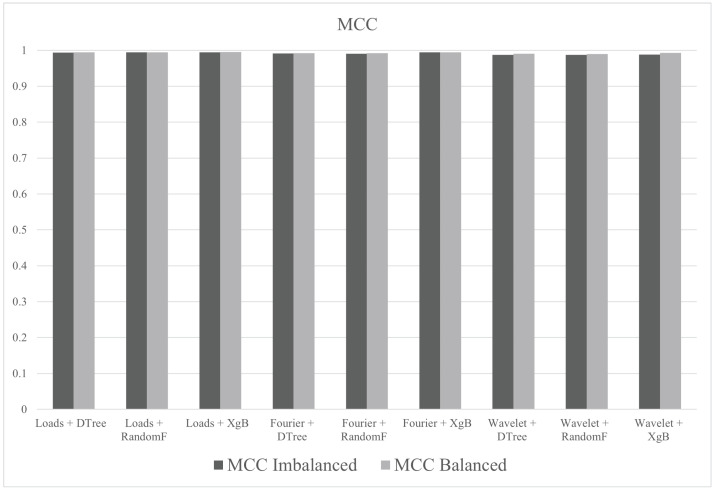
*MCC* for balanced and unbalanced datasets.

**Table 1 sensors-21-04546-t001:** Distribution of cards by type.

Type of Operation Modes and Automation Fails	Number of Cards
Normal	10,282
Gas interference	260
Fluid pound	38,298
Traveling valve leak	67
Standing valve leak	172
Rod parted	29
Gas lock	6
Tubing anchor malfunction	30
Sensor fault—rotated card	895
Sensor fault—line card	73

**Table 2 sensors-21-04546-t002:** Resources and tags used for each experimentation.

Resource	Tag
Fourier descriptors	Fourier
Wavelet descriptors	Wavelet
Load values	Loads
Decision tree	DTree
Random forest	RandomF
XGBoost	XgB
AutoML	AML
Balanced training dataset	Balanced
Imbalanced training dataset	Imbalanced
With hyperparameter tuning	WithTuning
No hyperparameter tuning	NoTuning

**Table 3 sensors-21-04546-t003:** Organization of tests in 4 groups. All tests were performed with two descriptors and the load values of the downhole card.

	Group A	Group B	Group C	Group D
ML algorithms	Decision tree, Random forest and XGBoost	AutoML	Decision tree, Random forest and XGBoost	Decision tree
Number of instances per class	30	30	90	180
Test set size	50,098	50,098	50,098	50,098
Number of tests	9	3	9	3
Maximum accuracy (%)	86.005	84.516	97.900	98.174

**Table 4 sensors-21-04546-t004:** Pipelines of group B.

	Pipeline
Loads + AML	RobustScaler [76] + KNeighborsClassifier [77]
Fourier + AML	GradientBoostingClassifier [78] + KNeighborsClassifier
Wavelet + AML	LinearSVC [79] + GaussianNB [80] + ExtraTreesClassifier [81]

**Table 5 sensors-21-04546-t005:** Training set organization.

Type Card	Total	Percentage
Normal	8225	20.52%
Gas interference	208	0.52%
Fluid pound	30,628	76.42%
Traveling valve leak	54	0.13%
Standing valve leak	137	0.34%
Rod parted	23	0.06%
Gas lock	5	0.01%
Tubing anchor malfunction	24	0.06%
Automation fail—rotated card	716	1.79%
Automation fail—line card	58	0.14%

**Table 6 sensors-21-04546-t006:** Organization of tests into 3 groups. All tests were performed with two descriptors and the load values.

	Group E	Group F	Group G
ML algorithms	Decision tree, random forest and XGBoost	Decision tree, random forest and XGBoost	AutoML
Training set state	Balanced and imbalanced	Balanced and imbalanced	Balanced and imbalanced
With or without tuning in?	No	Yes	Yes
Training set size	40,078	40,078	40,078
Test set size	10,020	10,020	10,020
Number of tests	18	12	6
Maximum accuracy (%)	99.81	99.82	99.84

**Table 7 sensors-21-04546-t007:** Pipelines of group G.

	Pipeline
Loads + AML + imbalanced	KNeighborsClassifier
Fourier + AML + imbalanced	Normalizer + KNeighborsClassifier
Wavelet + AML + imbalanced	Normalizer + KNeighborsClassifier
Loads + AML + balanced	RandomForest
Fourier + AML + balanced	MultinomialNB [85] + PCA [86] + KNeighborsClassifier
Wavelet + AML + balanced	BernoulliNB [87] + ExtraTreesClassifier

## Data Availability

The data present in this study are available on request from the J.N. author.

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
