# Peer review of "Diagnostic of Operation Conditions and Sensor Faults Using Machine Learning in Sucker-Rod Pumping Wells"

_sensors, 2021, doi:10.3390/s21134546_

Round 1

Reviewer 1 Report

An excellent prospective journal paper, which was well written and researched by the author's.

The subject discussed has considerable merit and both the conclusions and references were appropriate and valid, which is worth publication in the journal, illustrating the validity of the research into this investigation into 'pumping technology'.

Author Response

In file attach.

Reviewer 2 Report

This manuscript presents a machine learning solution to the diagnosis of operating conditions and sensor faults. It is well written, and the experiments are solid. However, a few comments need to be addressed before further consideration.

  1. Please highlight your contribution in the introduction. It seems that the current manuscript did not propose new algorithms.
  2. Please summarize the proposed framework/method as an algorithm block for better readability.
  3. The organization of the manuscript needs to be improved. For example, the discussion of the patterns for the operating conditions can be moved to experiments and dataset description.
  4. Please compare the proposed framework to state-of-the-art methods.

Author Response

In file attach.

Reviewer 3 Report

This might be an interesting paper concerning feature selection  -other works are more concerned with pattern recognition. I just have three remarks:

  1. In figure 22 (shouldn't be a table?) you quote 6 cases of gas lock perfectly classsified  (6 TP and 0 FP/FN). So how exactly did you obtain these results? Are they based on the training data set? Please be very specific on labeling source of the results: either from the training or test dataset?
  2. Groups A, B, C are labeled as 30 cases datasets yet in the text (line 548) you quote that each class has 30 cases. So what exactly are the numbers of cases in these groups? Are all the cases eqally represented? Please be more specific on the structure of the datasets
  3. Additionally, what's the point of iterating over 30 - 180 cases vs. 10 000 or 50 000 cases? Several orders of magnitude difference and after all you show that going into these ranges of datasets (40 000) is in fact a good solution and provides perfect results. I find it as an inconsistency in the design of your work.

Round 2

Reviewer 2 Report

The authors has addressed my comments, I have no further questions.

Reviewer 3 Report

Thank you for the clarifications and modifications of the text. It is a go now.